# Chloride Transport and Related Influencing Factors of Alkali-Activated Materials: A Review

**DOI:** 10.3390/ma16113979

**Published:** 2023-05-26

**Authors:** Xiaomei Wan, Yunzheng Cui, Zuquan Jin, Liyan Gao

**Affiliations:** 1School of Civil Engineering, Qingdao University of Technology, Qingdao 266520, China; cuiyunzheng1998@163.com; 2Collaborative Innovation Center of Engineering Construction and Safety in Shandong Blue Economic Zone, Qingdao 266033, China; 3School of Science, Qingdao University of Technology, Qingdao 266520, China; 32708gly@sina.com

**Keywords:** alkali-activated materials, chloride transport, chloride resistance, micro-structure, solidification of chloride, test method of chloride transport

## Abstract

Chloride transport is a vital issue in the research on the durability of alkali-activated materials (AAMs). Nevertheless, due to its miscellaneous types, complex mix proportions, and limitations in testing methods, the reports of different studies are numerous and vary greatly. Therefore, in order to promote the application and development of AAMs in chloride environments, this work systematically reviews the chloride transport behavior and mechanism, solidification of chloride, influencing factors, and test method of chloride transport of AAMs, along with conclusions regarding instructive insights to the chloride transport problem of AAMs in future work.

## 1. Introduction

Chloride-induced corrosion is one of the main reasons for durability degradation of reinforced concrete structures. The transport of chloride ions in concrete is directly related to the corrosion time of reinforcement; the related study is of great significance to improve the durability and prolong the service life of concrete structures [1]. Alkali-activated materials (AAMs) are a new type of building materials, which are generated by the reaction between pozzolanic active materials or hydraulicity aluminosilicate powders and alkaline activators [2]. The raw binding materials of AAMs, also called precursors, are generally industrial byproducts or industrial wastes. They not only have low production costs but also have special significance in the utilization of waste. Furthermore, AAMs yield higher environmental benefits, as shown in Figure 1, which implies that alkali-activated concrete can reduce CO_2_ emissions by more than 50% compared with ordinary Portland cement-based with the same strength grade [3]. In recent years, in order to yield higher environmental benefits, AAMs have increasingly favored the study of unconventional precursors and activators [4,5,6]. Except for the environmental impact, AAMs also have other excellent performance characteristics, such as higher early strength, comparable mechanical performance, and better resistance to sulfate attack and acid corrosion [7,8,9]. Therefore, AAMs seem to as the most potential alternatives to OPC.

Although AAMs have been demonstrated to have superior durability, there are inconsistent results in numerous studies on the chloride transport of AAMs. Mangat et al. [10] reported a higher diffusion rate of chloride in AAS concrete compared with OPC with a similar strength grade. Alkali-activated fly ash (AAFA) concrete seems to have little or even no application in chloride corrosion engineering [11]. However, the majority of studies reported good chloride resistivity of AAMs [12,13]. Hu [14] and Bondar et al. [15] observed that alkali-activated slag (AAS) has lower chloride permeability than OPC. Monticelli et al. [16] found that although AAFA mortar has a higher porosity than PC mortar, the chloride concentration on the steel bar surface is lower under the same immersion time. The reasons for these inconsistent results are related to different factors, such as the physicochemical properties of the precursor, type of activator, and mix proportions. Therefore, it is critical to understand the chloride transport mechanism and related influencing factors of AAMs.

For a better understanding of chloride transport in AAMs, the relationship between chloride transport performance and the micro-structure of AAMs is reviewed in this paper. The effect of chloride solidification of AAMs on chloride transport is also summarized. Some factors, including precursor, activator, mix proportions, and curing conditions, affecting the chloride transport performance of AAMs are reviewed. Furthermore, the differences in chloride transport results for AAMs due to test methods are indicated, and some ways to improve the accuracy of the test results of AAMs are presented. The points that need attention in future research work for advancing the application and development of AAMs in chloride salt environments are provided.

## 2. Chloride Transport and Micro-Structure of AAMs

It is generally believed that external chloride enters porous cementitious materials mainly through permeation under hydrostatic pressure, capillary absorption, and diffusion. In the presence of pressure on the hydraulic head, the migration of chloride will be accelerated into the cementitious materials with the flow of water, and the flow velocity can be calculated by Darcy’s law [17,18]. When unsaturated porous cementitious material is in contact with seawater, due to the presence of liquid surface tension within capillary pores, chloride will be rapidly absorbed into the material together with water. This effect is also known as the wicking effect [19,20]. When concrete is saturated sufficiently, chloride diffuses from the outside to the material under the derivation of internal and external concentration gradients [21]. Normally, the transport of chloride mainly depends on diffusion. The typical diffusion model is based on Fick’s second law, in which the chloride diffusion coefficient becomes a key index to evaluate the chloride transport performance of concrete [22,23].

For porous cementitious materials, the influence of different factors on chloride transport in cement-based materials can be studied based on internal structure, especially micro-structure. It is the same for the chloride transport in AAMs. By now, many studies have reported the hydration products, pore structure, interfacial transition zone, and pore solution of AAMs. In this section, the relationship between hydration products, pore structure, interfacial transition zone, pore solution, and chloride transport will be discussed.

### 2.1. Solid Phases Assemblage of Hydration Products of AAMs

X-ray diffraction (XRD), Fourier transform infrared spectroscopy (FTIR), nuclear magnetic resonance (NMR), and some other testing techniques can be applied to characterize the hydration products of cementitious materials effectively. It is demonstrated that the hydration products of AAMs and OPC are significantly different in chemical composition and structure. It is well known that the primary hydration product of OPC is calcium silicate hydrate (C-S-H), accounting for about 70%. Additionally, it also includes secondary products such as calcium hydroxide (CH), ettringite, and AFm phases. Depending on the composition of prime materials, AAMs are separated into two groups [24]. One is produced from low calcium aluminosilicate precursors, e.g., fly ash (FA) and metakaolin (MK). Silica and reactive alumina of precursors was dissolved in high pH activators and polymerization with Si-O-Al-O as the basic unit [25,26,27], formatting sodium aluminosilicate hydrate (N-A-S-H) gels with a zeolite-like three-dimensional framework structure [12,28]. In addition, crystalline or semi-crystalline zeolites are secondary products that have a microporous structure with a highly regular structure of pores and chambers [29]. This group is also named geopolymer due to its polymeric structure [24]. The second group is produced from high calcium aluminosilicate precursors, e.g., blast furnace slag (BFS), where calcium aluminosilicate hydrate (C-A-S-H) gels with a tobermorite-like structure [24] (Figure 2) are the primary hydration products. The layered double hydroxides (LDHs) are the secondary products [30,31] of the general formula [$M_{1}^{II}$ − *_x_*M^III^*_x_*(OH)_2_]*^x+^*[A*^m−^*]*_x/m_*·*n*H_2_O, which are a group of minerals that have a positively charged layered structure [32].

It was found that the micro-structure of the geopolymer based on N-A-S-H gels was looser than C-(A)-S-H binder gels. Figure 3 shows the micro-structure of alkali-activated slag/fly ash; it can be observed that with the increase of slag content, fewer pores were found, and the micro-structure became denser [33]. This is attributed to the increase in the number of C-A-S-H gels with space-filling ability [34], according to EDS results. Compared with N-A-S-H gels, whose morphology tends to crystallize, especially with high curing temperature and long curing time [24], C-A-S-H gels are mostly in an amorphous state and have significant amounts of bound water, which strikingly refines the pores of the system [35]. In general, the dense matrix structure can effectively hinder the transport of chloride ions and other corrosive species [36].

### 2.2. Pore Structure

Some simple classifications of pores were proposed according to the pore size. Young et al. [37] divided the pores of cementitious materials into three classes: gel pores (<10 nm), capillary pores (10~10,000 nm), and macropores (>10,000 nm). Mehta and Manmohan [38] broadly classified pores into four categories: <4.5 nm, 4.5~50 nm, 5~100 nm, and >100 nm. Some characteristic parameters are defined according to the pore size distribution, such as the pore size at half of the total pore volume is defined as the average pore diameter, and the most probable pore diameter is the one with the highest probability of existence. Technologies such as mercury intrusion porosimetry (MIP), N_2_ adsorption, and X-ray computed tomography (X-CT) can be used to obtain the pore size distribution and structure characteristics.

For cementitious materials, low porosity, small pore diameter, poor connectivity, and high tortuous degree are often related to low chloride permeability. Capillary pores are commonly considered to be the main channel of media transport. Hence, the less volume of capillary pores, the lower the chloride permeability of concrete [39]. By regression analysis of the chloride diffusion coefficient and different pore structure characteristic parameters of concrete, it is found that there is a significant linear relationship between chloride diffusion coefficients and pore structure parameters. Compared with porosity, the total specific pore volume, most probable pore diameter, and median diameter have a higher correlation with the chloride diffusion coefficient [40]. Based on the alternating-current (AC) impedance test, Wang et al. [41] proposed a method for characterizing the tortuosity and constrictivity of the connected pores to explore the influence of the chloride diffusion coefficient and found that the connected pores in cement-based materials play a decisive role in the control of chloride ion migration.

N-A-S-H gel-based AAMs generally have macropore structures accompanied by small gel pores and high porosity, while C-(A)-S-H gel-based AAMs have denser pore structures with lower porosity [42]. The results of Lloyd et al. [43] by Wood’s metal intrusion porosimetry showed that alkali-activated pastes exhibit lower porosities compared with OPC, and slag-based alkali-activated pastes have the lowest porosities (Figure 4). For an alkali-activated FA/GGBS (AAFS) binary system, Babaee et al. [44] found that most AAFSs have finer pore distributions and lower porosity compared with OPC, and as the replacement of slag increases, the pore distribution moves gradually from capillary pores to gel pores. This is attributed to the generation of more C-A-S-H gels to fill pores. By increasing the alkali content or silicate modulus of activators, Hu et al. [14] found that the chloride migration coefficient of AAFS mortar decreases with the volume of pores of size 10–10^4^ nm, but the changes of gel pores less than 10 nm and pore larger than 10^4^ nm are not significant. Hence, the authors concluded that the chloride permeability of AAFS is mainly affected by the capillary pores.

### 2.3. Interfacial Transition Zone

In Portland cement concrete, compared with paste matrix, there is a higher content of calcium hydroxide crystals and ettringite in the interfacial transition zone (ITZ). Consequently, ITZ has a higher porosity than the paste matrix and would provide an easier pathway for chloride penetration into the interior [45]. However, the ITZ of AAMs has totally different features compared with PC concrete. It was found that the main product in AAFA concrete is N-A-S-H gels [46], and the outer products of ITZ in AAS are mainly C-A-S-H gels with a low Ca/Si ratio instead of large crystallites; as a result, the ITZ of AAMs shows a lower porosity than PC concrete.

Figure 5 presents the SEM micrographs of the OPC boundary and geopolymer concrete, where the ITZ of OPC is very easy to locate because of the obvious micro-crack-like structure along the edge of the aggregate, while the ITZ of the geopolymer is denser and more difficult to locate [47]. Fang et al. [48] investigated the evolution of ITZ in alkali-activated fly ash-slag concrete and found that, at the initial stage of the reaction, the main product in ITZ is C-(N)-A-S-H gels with low Ca content. Then, with the release of more Ca from the paste matrix, more C-(N)-A-S-H gels with high Ca content were gradually generated, with even lower porosity than the paste matrix after 28 d. As mentioned above, ITZ in alkali-activated concrete has a compact and dense micro-structure, which means that this region will not be a preferred pathway of transport for chloride and other aggressive ions [49,50].

### 2.4. The Composition of the Pore Solution

The pore solution of hardened cementitious materials is normally extracted by the steel-die method [51], and the chemical composition is often analyzed through inductively coupled plasma-optical emission spectroscopy (ICP-OES) and ion chromatography (IC) [52]. There are large differences between the pore solution composition of OPC and AAMs [51,53].

Table 1 presents the main element concentration of the pore solution of AAMs and OPC pasters at 28 d. It is obvious that the concentration of Na, Al, and Si elements in AAMs is tens or even hundreds of times higher than OPC, but the concentration of K element is ten or even hundreds of times lower than OPC. The AAS samples have the lowest Ca content; this could be attributed to the common ion effect [51]. The pore solutions had high concentrations of Si, OH^−^, and Na^+^; those ions might combine with Ca^2+^ to form solid reaction products and thus lower the Ca concentration in the pore solution [51]. Furthermore, the pore solution of AAMs contains a high concentration of S, mainly present in the forms of HS^−^ [31]. Previous researchers found that pore solutions have impacts on chloride diffusion [54]. Generally, cations (e.g., Ca^2+^ and Na^+^) have lower diffusion rates than chloride; to conserve the local electroneutrality, chloride slows down the diffusion rate [55]. Consequently, the high cation concentration of pore solution in AAMs reduces the chloride diffusion rate. The pore solution in AAMs also has a high OH^−^ concentration due to the use of an activator solution [56]. It affects the chloride solidification effect of AAMs, which will be discussed in the next section.

## 3. Chloride Solidification in AAMs

Chloride solidification is another important factor influencing chloride transport. When chloride penetrates into concretes, a proportion of the chloride ions are captured by hydration products leading to a decrease in the penetration rate [57] and solidification of chloride, including physical adsorption and chemical binding.

### 3.1. Physical Adsorption

Some studies have found that physical adsorption is the main chloride-solidified method for AAMs [52,58]. The surfaces of hydration products usually carry charges; ions (e.g., Ca^2+^, Cl^−^) of pore solution under the action of intermolecular force (e.g., Van Der Waals and electrostatic) adsorb to the surface of hydration products to compensate for the surface charge [59]. The surface charge of hydration products and the ions in the pore solution constitute the so-called electrical double layer.

As the main product in the geopolymer, N-A-S-H gels have been found to possess high physical adsorption capacity due to their large surface area [60]. Lee et al. [61] noted that the chloride binding capacity increases with the amount of N-A-S-H gel in alkali-activated slag/fly ash, demonstrating that N-A-S-H gels have higher chloride binding capacity than C-A-S-H gels. Additionally, it is well known that zeolites are an absorbent of several ions and molecules [29], while geopolymers are part zeolite.

More studies should focus on the chloride binding of AAS rather than the geopolymer. Zhu et al. [52] found that AAS exhibits a greater chloride binding capacity than OPC (Figure 6). The higher chloride binding capacity of AAS is related to the production of C-A-S-H gels and layered double hydroxides (LDHs) [62,63]. Cai et al. [59] synthesized the main hydration products of AAS with different compositions. The physical, chemical, and total chloride solidified amounts are shown in Figure 7. It is obvious that C-A-S-H gels exhibit higher chloride binding capacity than C-S-H gels and the chloride binding amounts increase with the increase of the Al/Si ratio. According to the literature reports [59,64,65], replacing Si in the C-S-H bridging with Al increases the gel layer spacing and specific surface area, increasing the Zeta potential; this provides more sites for the adsorption of chloride ions. Furthermore, LDHs possess greater chloride solidification capacity compared with C-(A)-S-H gels; this is attributed to the higher specific surface area and positive Zeta potential.

### 3.2. Chemical Binding

In ordinary Portland concrete, chemical binding normally refers to chloride’s reaction with the AFm-phase and the formation of Friedel’s salts and Kuzel’s salts [66]. Since there is no C_3_A and C_4_AF in alkali-activated binders, the mechanism of chloride chemical binding in AAMs is different from OPC. By adding alkali-activated fly ash mortars into NaCl solution for 2 years, Ismail et al. [12,67] only found the NaCl crystalline phases without new chlorine salt crystal phase observed from XRD results. Brough et al. [68] used an activator with a high NaCl content (by 8% weight of slag) to activate slag. After hydration for 28 days, the XRD results showed no significant AFm or Friedel’s salt peaks in the 8 ± 13° 2θ region, which is where Friedel’s salt is expected to appear.

Meanwhile, Zhu et al. [52] observed Friedel’s salt with a flake hexagon from SEM images of alkali-activated slag after exposure to chloride. Some studies have suggested that the chemical binding is mainly related to Layered double hydroxides (LDHs). There are two types of important LDH products in AAS. One is Mg–Al LDHs with a hydrotalcite-like structure, as shown in Figure 8, which bind chloride by interlayer ion exchange in addition to the surface adsorption of chloride [69]. The other is Ca–Al LDHs with hydrocalumite-like structure, also referred to as AFm phases [32], which can react with chloride to generate AFm-Cl with a similar structure to Friedel’s salt [70]. Ke et al. used sodium carbonate solution to activate slag, and observed two types of Friedel’s salt crystalline phases, (^R^AFm-(${CO}_{3}^{2-}$,Cl^−^)), which is close to the structure of rhombohedral hydrocalumite, and (^M^AFm-(${CO}_{3}^{2-}$,Cl^−^)), which is close to monoclinic hydrocalumite [71]. Zhang et al. [62] investigated the chloride binding of alkali-activated fly ash/slag, and found that Friedel’s salt only exists in AAS samples after physical adsorption in equilibrium.

Some anions (e.g., OH^−^, ${CO}_{3}^{2-}$, and ${SO}_{4}^{2-}$) in pore solution could affect chloride solidification capacity [72]. For AAS pastes immersed into NaCl solution with different pH values, with the increase of pH value from 11.0 to 13.5, the bound chloride amounts significantly decrease [52]. It is due to the competition between OH^−^ ions and Cl^−^ ions in pore solution. With the increase of OH^−^ ions in solution, more adsorption sites are occupied by the OH^−^ ion [52]. The Zeta potential of hydration products with different chemical compositions in neutral ultrapure water or NaOH solution is shown in Figure 9 [59]. The C-A-S-H gels have more negative Zeta potential in alkaline solution than ultrapure water—the Zeta potential of LDHs even turns from positive to negative. It leads to higher electrostatic repulsion forces in hydration products and reduced chloride solidification capacity of AAS. Other researchers reported [73,74,75] the bound chloride of LDHs by interlayer ion exchange can be replaced by anion with a stronger affinity, such as OH^−^, ${CO}_{3}^{2-}$, and ${SO}_{4}^{2-}$. Chen and Ye [76] investigated the impact of ${SO}_{4}^{2-}$ on the chloride diffusion in AAS and found that the characteristic peak of sulfate-hydrotalcite becomes broader and more asymmetric with increasing ${SO}_{4}^{2-}$ concentration, supporting the competition of sulfate for intercalation reducing the chloride binding capacity of hydrotalcite in AAS.

## 4. Factors Affecting Chloride Transport in AAMs

### 4.1. Precursors

The common precursors of AAMs involve blast furnace slag, fly ash, and metakaolin. There are significant differences in the chemical composition characteristics of various precursors, which are demonstrated as the main factors affecting the chloride transport in AAMs. Osio-Norgaard et al. [42] summarized the average chemical composition of precursors from the literature concerning chloride transport in AAMs (see Figure 10).

Blast furnace slag is a byproduct of the steel industry. Compared with other precursors, slag has higher CaO and MgO contents, as shown in Figure 10. Additionally, most oxide in slag belongs to vitreous bodies with high reactivity [77]. Attributing to the high CaO contents and reaction activity, AAS produces more C-A-S-H gels to refine pore structure and leads to higher chloride resistance compared with geopolymers [78]. With replacing fly ash with slag, the MIP results showed that the total porosity decreased, and the critical pore size decreased from 8 to 3 nm, which is about two orders of magnitude lower than the critical pore size of pure fly ash (~0.15 μm) [33]. Zhang et al. [79] measured the chloride diffusion coefficient of alkali-activated fly ash/slag concrete after being immersed in NaCl solution for 90 days by a nature diffusion test, with the content of slag increased from 30% to 100%, chloride diffusion coefficient decreased from 1.78 × 10^−12^ m^2^/s to 0.48 × 10^−12^ m^2^/s, and proved the effect of slag to improve the chloride penetration resistance of AAMs. The result is consistent with research on the positive effect of slag addition in terms of reduction in chloride penetration [78,80].

In addition to CaO, other oxides in slag also have a great influence on the hydration products of AAS. The content of Mg in slag relates to the formation of hydrotalcite-type phases [81]. Mg–Al LDHs possess a strong chloride solidification capacity and a better-refined pore structure [82,83]. Yoon et al. [84] blended MgO to replace fly ash and slag, and found that the replacement of 5% (weight) increases the formation of hydrotalcite, reduces the total porosity, and improves the chloride penetration resistance. However, with further replacement of MgO to 10%, the ponding test results revealed the chloride penetration even deeper than that without MgO; this may be related to the excess MgO increasing the formation of weak and expansive brucite [85]. Hence, appropriately increasing the content of MgO will benefit the chloride penetration resistance. For Al_2_O_3_, HaHa et al. [86] reported the improved formation of C-A-S-H gels and hydrotalcite with the increase of Al_2_O_3_ content in slag, with a lower Mg/Al ratio. Although the porosity has no significant variation, more C-A-S-H gels have higher Al/Si and hydrotalcite with lower Mg/Al, which increase the chloride binding capacity of AAS [59]. Yang et al. [87] partially replaced the slag with Ca(OH)_2_ and nano-Al_2_O_3_ and found a significant increase in the chloride binding capacity of AAS, which shows that the initial Ca/Al ratios of the precursors are an important factor affecting the chloride binding capacity.

Fly ash possesses lower reaction activity in comparison with slag due to its more crystalline mineral phases. Therefore, AAFA normally needs curing at high temperatures to obtain higher bulk properties [88]. Contrary to slag, the low calcium content and reactivity make AAFA have a loose micro-structure, and the overall chloride penetration resistance is weaker than AAS. Gluth et al. [89] evaluated the chloride permeability of concretes of AAFA and AAS by accelerated chloride penetration testing and rapid chloride migration testing. The results suggested the chloride migration coefficients of AAFA were higher than AAS by about two orders of magnitude.

Some studies have tried to utilize fly ash with high calcium content to improve chloride resistance while obtaining contrary results to what was expected [90,91]. Winnefeld et al. [92] investigated the hydration products and micro-structure of AAFA manufactured by the fly ash with various calcium contents and found the examined low calcium AAFA produces more C-(N)-A-S-H gels and a more compact micro-structure than high-calcium AAFA. The content of the vitreous phase in high-calcium fly ash (about 10–30%) is lower than in low-calcium fly ash (more than 60%); hence, the reactivity of high-calcium fly ash is lower. Kupwade et al. [93] tested the chloride diffusion coefficient of AAFA prepared with Class F fly ash (1.97% CaO and 5.00% CaO content) and Class C (12.93% CaO content) according to ASTM C1556; the results indicated that the specimens manufactured by Class C fly ash possess the highest chloride penetration rate. According to the report of Yip et al. [94], the formation of C-S-H gels together with the N-A-S-H gels occurs only in a system at low alkalinity. In the presence of high concentrations of NaOH, the coexistence of the two phases is not observed unless there is a substantial amount of a reactive calcium source in the precursors. Therefore, in the study of Kupwade et al. [93], the utilization of high NaOH concentrations and lack of adequate amounts of reactive calcium reduces the formation of C-A-S-H gels in alkali-activated Class C fly ash; as mentioned above, lower C-A-S-H gels lead the system to a higher chloride permeability.

Metakaolin is one type of calcined clay, and due to the low CaO content, alkali-activated metakaolin (AAMK) possesses a poor pore structure and low chloride penetration resistance. Studies have shown AAMK and AAFA with similar chloride migration coefficients, which are approximately two orders of magnitude higher than AAS [89,95]. However, as supplementary cementing materials, they effectively improve mechanical properties and durability due to their high reactivity and pure aluminosilicate [96,97].

Fineness is another important physical property of precursors. The reactivity of precursors increases with the particle fineness, which improves the level of hydration and obtains a more compact pore structure and lower chloride permeability [42]. Nath et al. [98,99] found that with the reduction of particle size of fly ash/metakaolin, the number of unreacted and non-bridging particles in the geopolymer decreases, the micro-structure becomes more uniform, and the porosity decreases significantly, which is related to better chloride resistance. It should be noted that finer slag particles with larger surface area and higher reactivity usually have higher water demand; insufficient water content may lead to inadequate reaction and low chloride resistance [100,101]. Hence, it is still challenging to study the relationship between water content and precursor fineness.

### 4.2. Activators

Compared with the hardening of Portland cement caused by the introduction of water, the effect of using an activator solution with different alkaline components on the alkali-activated binder is more complex [24]. According to the chemical composition of alkali activators, it was divided into several categories, among which alkali metal hydroxide (e.g., NaOH and KOH) and silicate solution (e.g., Na_2_SiO_3_) are the most commonly used activator because they are easy to obtain and have excellent activation effect [102]. Compared with silicate activation, the reaction degree of the binder activated by hydroxide before hardening is generally lower and usually forms worse pore structures [14,103]. In addition, it was found that when NaOH was used to activate slag, with the increase of age, calcium in C-S-H gels was replaced by sodium to form C-N-S-H gels [104]. It was reported that the removal of divalent cations would cause greater damage to the glass structure than the removal of monovalent cations [105]. Therefore, the worse pore structure is generated during later hydration, which also impairs the chloride resistance. On the contrary, AAMs generally show high chloride penetration resistance when Na_2_SiO_3_ is used as the activator—this is attributed to the presence of more ${SiO}_{4}^{4-}$, which reacts with the initial release of Al to form more aluminosilicate oligomers or reacts with Ca^2+^ to form C-S-H with a low Ca/Si ratio [105].

In recent years, green alkali activators such as sodium carbonate solution have received more and more attention. Due to the low alkalinity, when sodium carbonate is used as an activator, the precursor is more difficult to dissolve, so the pore structure and chloride permeability resistance of sodium carbonate-activated materials are worse than those of NaOH or activated Na_2_SiO_3_ [71,103]. Jiao et al. [106] reported that the micro-structure of AAS mortar will become denser, and the volume of mesopores will decrease when sodium carbonate is properly added to NaOH. The existence of carbonate compounds formed by ${CO}_{3}^{2-}$ anions will lead to fine pore structures [107]. It has been confirmed that small amounts of substituted NaOH with sodium carbonate or increasing the concentration of sodium carbonate solution can enhance the resistance to chloride penetration [108]. Through SEM analysis, it is confirmed that lower permeability is due to the formation of a large amount of calcite and a lower area fraction of unreacted slag particles.

### 4.3. Mix Proportion

#### 4.3.1. Alkali Content

The reaction of AAMs is the same as other chemical reactions that follow the kinetic law [51]. Higher concentrations of OH^−^ in the activator will accelerate the dissolution of the aluminosilicate precursor and release more chemical elements, increasing the formation of hydration products [109]. In alkali-activated systems, the alkali content is usually defined as the molar concentration of the solution or the percentage of metal oxide in the weight of the precursor [67]. Numerous studies have demonstrated that the increase of alkali content enhances the chloride penetration resistance due to lower porosity and finer pore structure [14,110,111,112]. The report of Chindaprasirt et al. [112] suggested that with the increase in NaOH concentration from 8 to 18 molar, AAFA concrete exhibited lower chloride permeability due to the denser micro-structure and higher chloride binding capacity.

However, the opposite conclusion also exists, with the increase of alkali content leading to an increase in chloride permeability [67,113,114]. Fang et al. [113] observed that the drying shrinkage of AAS mortars gradually increases with alkali contents, leading to microcracks, which is related to the high chloride permeability of AAS. According to the reviewed research results, AAMs with 6–9% alkali content generally show low chloride permeability [14,67,110,111,112,113,114].

#### 4.3.2. Water Content

In cement-based material systems, the ratio of water to cementitious binder (W/B) is an important factor affecting bulk properties. In some literature on AAMs, it can also be expressed as the ratio of alkali solution to cement-based binder (S/B) or the ratio of total liquid to total solid (L/S). It is generally accepted that decreasing the W/B can effectively refine the pore structure and reduce porosity, and this enhances the resistance to chloride penetration. Zhang et al. [79] investigated the effect of different W/B (0.40, 0.45, and 0.50) on the chloride resistance of alkali-activated fly ash/slag, and found that with the decrease of W/B, the volume of capillary pores decreases significantly (Figure 11), and the natural diffusion coefficient of chloride decreases from 3.2 × 10^−12^ m^2^/s to 1.1 × 10^−12^ m^2^/s. On the premise of proper workability, Zhu et al. [115] found that chloride penetrates faster in AAFA paste and mortar with an increase of L/S in the range of 0.6, 0.7, and 0.8. Najimi et al. [114] found that by reducing the S/B of alkali-activated natural pozzolan/slag mortars from 0.60 to 0.56, the water absorption decreased by about 8%, and the chloride penetration depth decreased by about 32.5%.

However, blindly reducing the water content is not conducive to the resistance of the chloride permeability of concrete. The insufficient water content will make the cementitious material unable to react fully and reduce the production of hydration products [116]. Additionally, the reduced water content leads to an increase in viscosity in the mixture and captures a large amount of air that cannot be expelled; thus, more large voids are produced [117]. To obtain the best chloride resistance, the water content needs to be generally limited to a W/B (S/B or L/S) from 0.4 to 0.5 [79,114,115,117].

#### 4.3.3. Silica Content

In alkali-activated systems, the main source of silica is the precursors. A large number of studies have shown that more available silica content in the alkali-activated system promotes the formation of hydration products, refines the pore structure, and obtains larger chloride resistance. For example, Ma et al. [111] proved that with the increase in the concentration of SiO_2_, the gel pore structure of AAFA paste was finer, and the measured water permeability was reduced by about 1–2 orders of magnitude. Hu et al. [14] found that the increase in silicate modulus improved the chloride binding ability of AAS concrete, thus slowing down the migration of chloride. Other substances rich in silica show significant improvement in chloride resistance as well. Ramezanianpour et al. [118] verified that adding 2% nano-silica can reduce the chloride migration coefficient by 28% and 21% at 28 and 90 days, respectively. This is because the high pozzolanic reactivity and the physical filler effect of nano-silica significantly improve the micro-structure and properties of the AAS mixture.

However, the high silicate modulus is not always conducive to the resistance of AAMs to chloride penetration. Ma et al. [119] found that with the increase of silicate modulus from 0.75 to 1.50, the chloride diffusion coefficient of AAS concrete in the unsteady chloride diffusion test decreases significantly. However, when the silicate modulus is further increased to 2, the chloride diffusion coefficient is even higher than that of the sample with a silicate modulus of 0.75, as shown in Figure 12. One explanation is that too high of a silicate modulus leads to an increase in drying shrinkage and then generates more cracks to facilitate chloride transport. Zhang et al. [120] found that the samples activated by water glass with a silicate modulus of 1.8 have a larger chloride diffusion coefficient than those activated by NaOH (can be considered modulus 0). They observed from SEM (Figure 13) that the former has obvious cracks. This proved that an excessive silicate modulus increases the chloride transport in AAMs. Therefore, a silicate modulus between 1 and 1.5 is probably the best choice when silicate solution is the activator.

### 4.4. Curing Conditions

Curing conditions (e.g., temperature and humidity) also have a significant impact on the reaction of alkali-activated systems. Curing at high temperatures can accelerate the dissolution of precursors and promote the generation of more products. Zuo et al. [51] extracted the pore solution of AAFA and analyzed it according to thermodynamics. It was found that the temperature increase from 40 °C to 60 °C results in more hydration products with higher cross-linking degrees and a more stable structure. Noushini et al. [121] reported that compared with the ambient-cured fly ash-based geopolymer, heat curing effectively reduces the volume of permeable pores and obtains higher resistivity. In the AAS system, studies suggested that high-temperature curing tends to increase the crystallinity and promote the formation of C-A-S-H gels, which leads to a dense micro-structure, thus enhancing the resistance to chloride penetration [109,122].

The improvement of the humidity environment also promotes the dynamics and degree of the hydration reaction, increasing the chloride resistance [42]. For example, Chi et al. [123] compared the effect of different humidity conditions on the chloride permeability of AAS concrete, including curing in air, under saturated limewater, and in a curing room with a relative humidity of 80% RH. The chloride rapid penetration test results show that curing with a relative humidity of 80% RH has the lowest total charge passing value. Mangat et al. [124] suggested that the alkali-activated slag material cured under wet/dry curing (placing the sample in the air for 3 days, then immersing it in water for 24 days) is the best curing regime. Studies [125] confirmed that curing in saline water could reduce the amount of unreacted particles and pores in fly ash-based systems, and even the penetration of chloride into the matrix was found to be negligible.

### 4.5. Other Factors from the Environment

Carbonation has an impact on the long-term performance of Portland cement-based concrete in the atmospheric environment. The acidic CO_2_ dissolves in the pore solution and reacts with calcium-rich hydration products, causing microstructural damage to the matrix. [126]. In AAMs systems, because there is no Ca(OH)_2_ present, Ca^2+^ in C-(A)-S-H gels become the main target for reaction with the carbonate. The pore structure of AAMs tends to be more damaged by carbonation, and the degradation of the micro-structure leads to higher chloride permeability. Bernal et al. [127] proved that at 1% CO_2_ concentration, the volume of AAS permeable pores gradually increased throughout the exposure time. Chaparo et al. [128] found that after carbonation for 45 days, the permeable void and water absorption of both AAS and OPC concrete increased, which had a good correlation with high chloride permeability. The deterioration of AAFS specimens resulting from carbonation is less serious than AAS due to the formation of C-(N)-A-S-H gels with a higher degree of cross-linking and polymerization; therefore, it had higher long-term performance under the combined environment of chloride and carbonation [129].

In the actual service environment, concrete is usually damaged by mechanical load, and the transportation of chloride is thus affected [130]. Numerous studies have been carried out on the chloride migration of OPC under mechanical load. The conclusions can be summarized as follows: under low compressive load, the chloride permeability decreased as a result of the closure of initial microcracks [131], while under tensile load, bending [132], or high compressive load, the chloride transport rate increases due to crack growth [133]. The research on the influence of mechanical load on chloride transport in AAMs systems is limited, but the results are generally consistent with OPC. Zhang et al. [120] pointed out that when the compressive load level is lower than 35% of the ultimate load, the chloride diffusion coefficient of water glass-activated slag decreases, and when the load level is higher than this threshold, the chloride diffusion coefficient increases with the increase of load.

## 5. Test Method of Chloride Transport Performance

The rapid chloride permeability test (RCPT) specified in ASTM C1202, rapid chloride migration (RCM) test specified in NT-Build 492, and bulk diffusion test specified in NT-Build 443 are widely used to detect chloride ion permeability in AAMs. Both RCPT and RCM are rapid electrical evaluation methods. The former measures chloride migration or conductivity, and the latter measures chloride penetration depth.

It is generally accepted that the applied high voltage and associated temperature rise will cause structural damage to tested samples. In addition, the test results are not only dependent on the connectivity and tortuosity of the network but also related to the ion concentration of the pore solution [134]. Therefore, it may not be able to reliably assess the chloride permeability of Portland cement containing supplementary cementitious materials or alkali-activated materials. Balcikanli et al. [135] and Shi et al. [103] proposed that the results of the RCPT test mainly depend on the pore solution chemistry rather than the pore structure, which can prove that the samples activated by Na_2_SiO_3_ have a denser pore structure and lower water permeability than the ones activated by NaOH, while the RCPT test results show that the passed charge of the former is higher.

RCM is maybe more suitable for evaluating the chloride permeability of alkali-activated concrete [14,89]. This may be attributed to the lower applied potential, longer test time, and less pore solution effect [11]. However, reports of RCM application to AAMs are not entirely positive either. Gluth et al. [89] found that for AAFA and AAMK, the result of the RCM test was consistent with the ponding test, but not for AAS. One reason is that for concrete with low chloride permeability like AAS, the depth of penetration is often small, and the bias caused by manual measurements is significant. It has also been reported that due to the high alkalinity of the pore solution of AAMs, the chloride ion concentration at the color change boundary of RCM cannot be determined accurately [136,137].

Compared with RCPT and RCM, the bulk diffusion test is undoubtedly a more accurate and reasonable method to evaluate the chloride permeability of AAMs. Of course, similar methods of chloride exposure, such as bulk diffusion tests, also face the problem of long testing periods and consuming manpower (after the immersion is completed, samples need layer grinding for chloride titration) [138].

Some studies reported different modified test methods for avoiding the shortcomings mentioned above. Noushini et al. [138] utilized the modified RCPT, using 10 V instead of 60 V to measure the charges passed through the geopolymer concrete, and the results suggested a good correlation with the method of ASTM C1556. In the traditional rapid chloride migration test, the chloride concentration at the color change boundary is usually taken as 0.07 M when calculating the chloride ion diffusion coefficient. Mases et al. [139] proposed that for AAS concrete, the color change value within the range of 0.13–0.45 M chloride concentration is more accurate.

## 6. Conclusions

This work reviews research on the chloride transport performance and related influencing factors of alkali-activated materials for promoting the application and development of AAMs in chloride environments. Based on the literature review above, the following conclusions have been drawn:

The chloride resistance of AAMs is dependent on the micro-structure. N-A-S-H gel-based AAMs normally exhibit a loose gel matrix and poor pore structure, while C-A-S-H gel-based AAMs, with relatively compact pore structures, exhibit higher chloride transport resistance. ITZ in AAMs is denser than OPC, which greatly reduces the rate of chloride transport in this region. High alkali metal cation concentrations in the pore solution of AAMs decrease the chloride transport rate of AAMs more than OPC, but this still needs further investigation.

Physical adsorption is the main mode of chloride solidification in AAMs, which is attributed to the large specific surface area and more positive Zeta potential of hydration products. A reasonable increase of LDH phase formation to increase the chloride resistance of AAMs needs to be explored.

Among the factors influencing the chloride transport of AAMs, the physicochemical properties of the precursors are the most critical, especially the content of reactive CaO. The research and application of green activators should not be neglected as well. Lowering the water–cement ratio and increasing the alkali content can reduce the chloride transport rate in AAMs, but it needs to be controlled within a reasonable range. Suitable curing conditions (high temperature and enough humidity) can further enhance the resistance to chloride transport in AAMs.

The high ion concentration in pore solutions causes inconsistent results when current accelerated methods were used to test the chloride transport performance of AAMs. Further improving the experimental methods and modification standards is still needed for more conveniently and accurately evaluating the chloride transport performance of AAMs.

## Figures and Tables

**Figure 1 materials-16-03979-f001:**
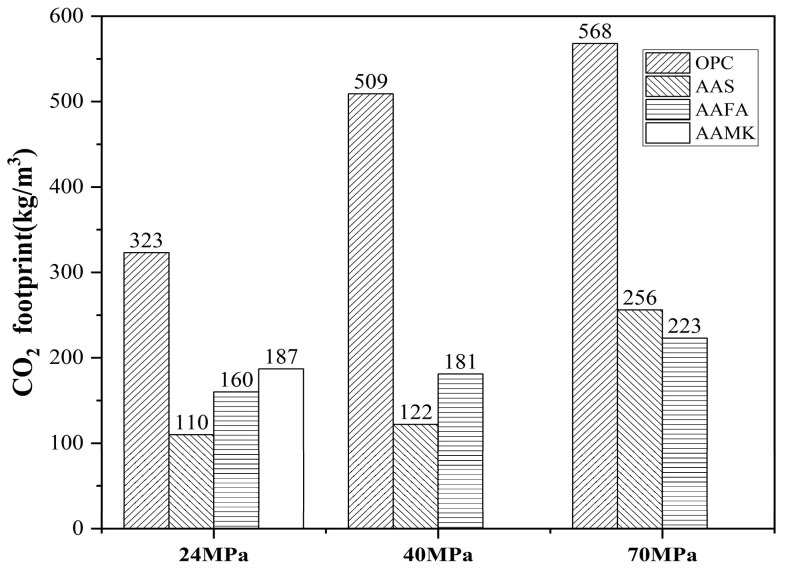
CO_2_ footprint for different concrete types [3].

**Figure 2 materials-16-03979-f002:**
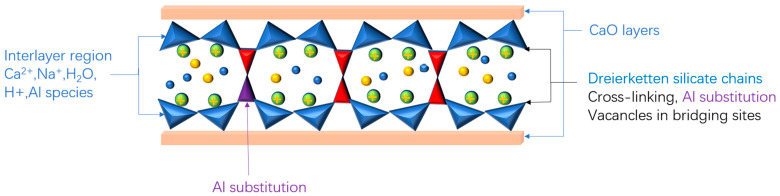
Tobermorite-like C-A-S-H gel structure model [24].

**Figure 3 materials-16-03979-f003:**
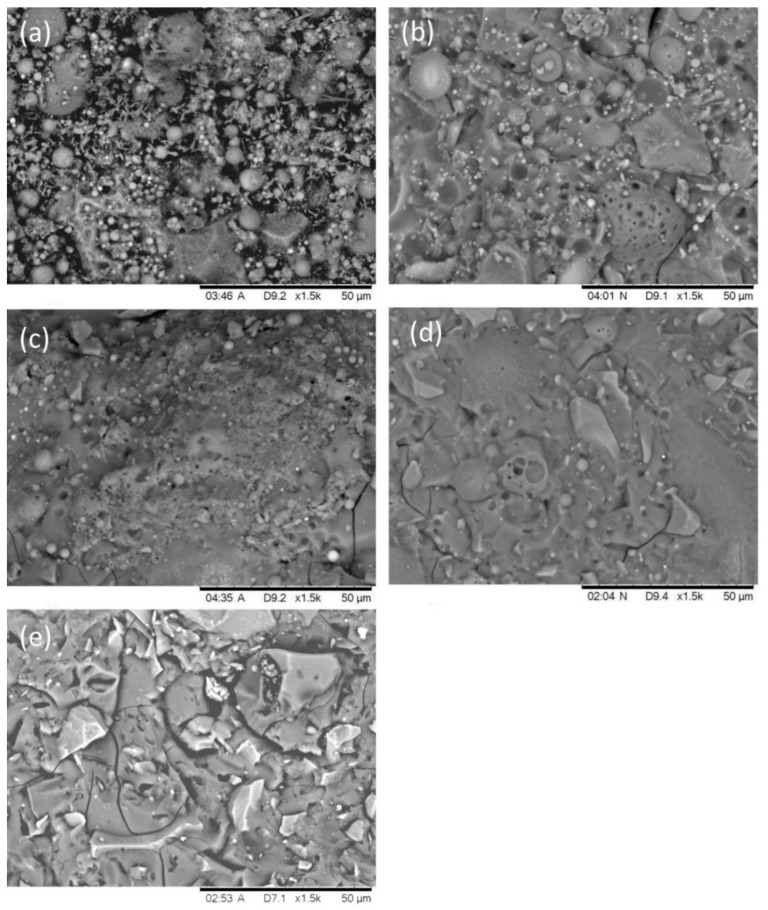
The micro-structure of the AAFS with the slag content of (**a**) 0%, (**b**) 30%, (**c**) 50%, (**d**) 70%, and (**e**) 100% at 7 days [33].

**Figure 4 materials-16-03979-f004:**
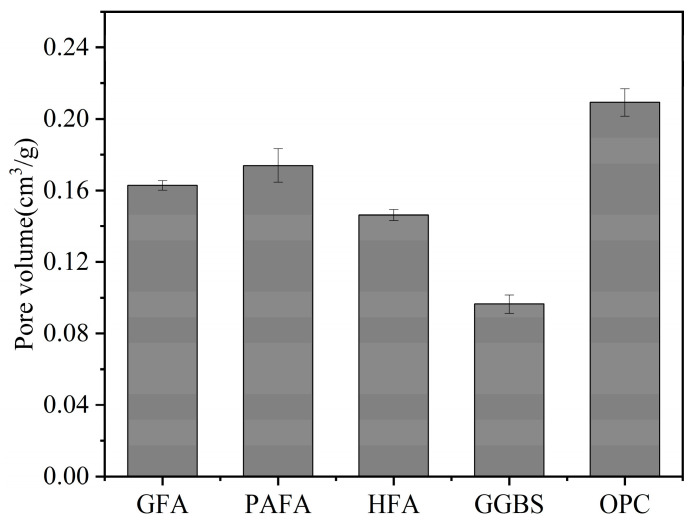
The porosities of inorganic polymer cements synthesized with different binder materials compared to OPC [43].

**Figure 5 materials-16-03979-f005:**
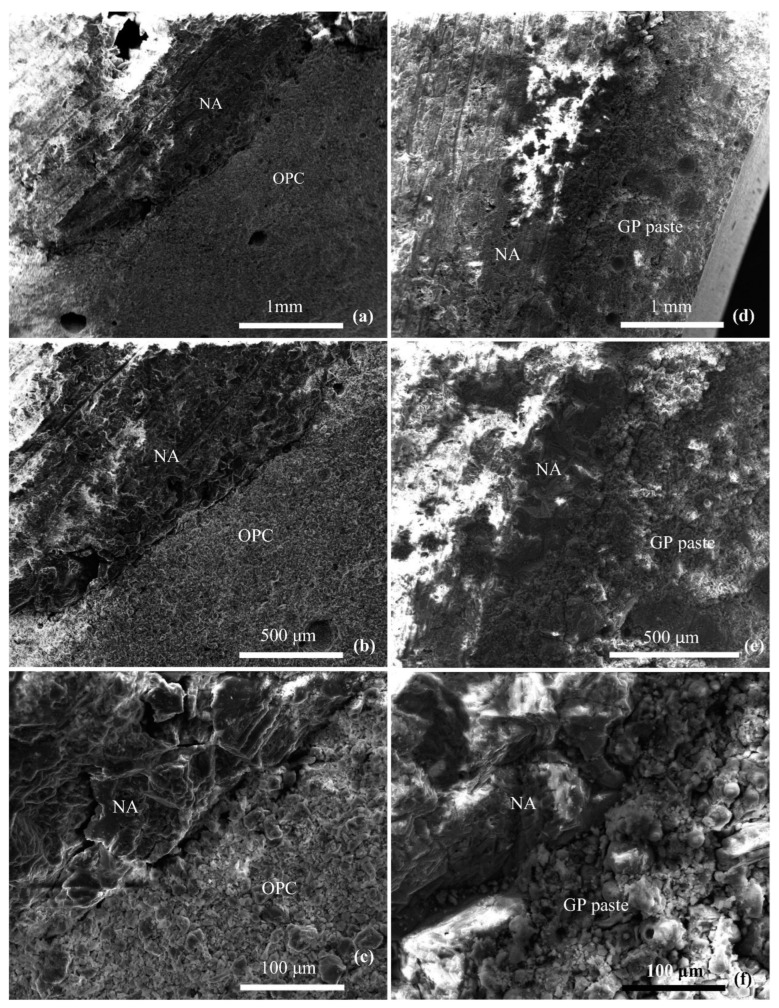
SEM micrographs of interfacial transition zones: (**a**–**c**): OPC-NA ITZ at different resolutions and (**d**–**f**): GP-NA ITZ at different resolutions. NA = natural aggregate, OPC = ordinary Portland cement, GP = geopolymer [47].

**Figure 6 materials-16-03979-f006:**
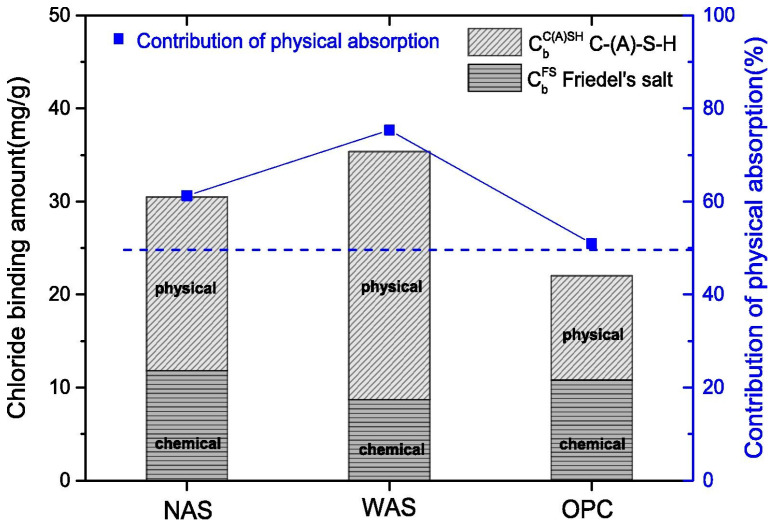
Amounts of physically adsorbed chloride and chemically bound chloride of NAS (NaOH-activated slag), WAS (water glass-activated slag), and OPC [52].

**Figure 7 materials-16-03979-f007:**
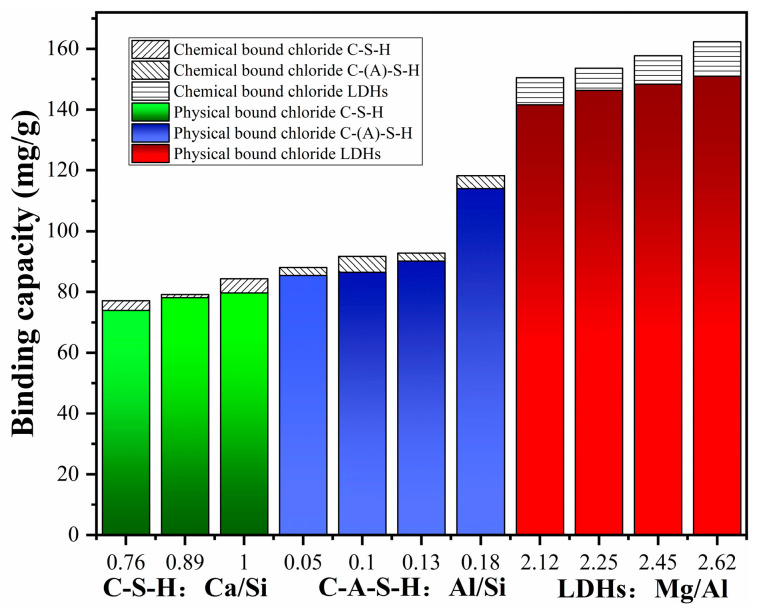
The maximum physical, chemical, and total chloride binding amount of different synthesized products in neutral (pH = 7) NaCl solution [59].

**Figure 8 materials-16-03979-f008:**
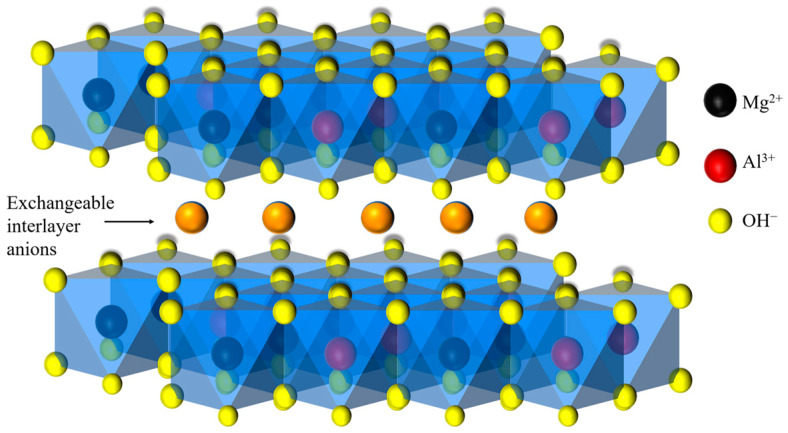
Schematic diagram of Mg–Al layered double hydroxide structure.

**Figure 9 materials-16-03979-f009:**
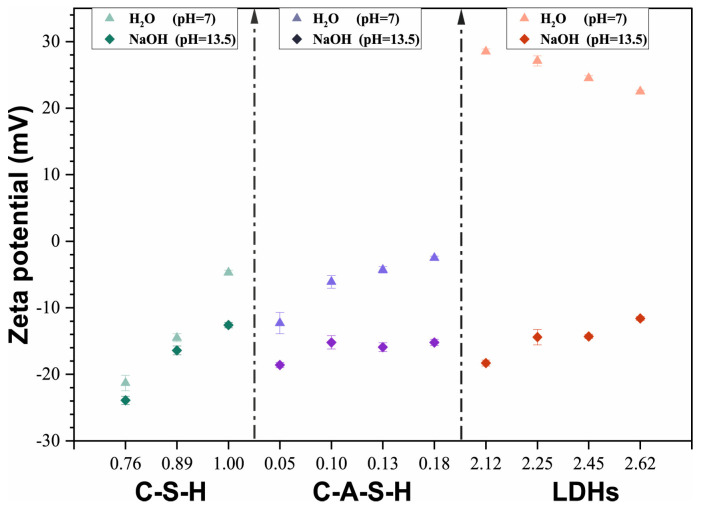
Zeta potential of the synthesized C-S-H gels, C-A-S-H gels, and LDHs with different chemical compositions in neutral ultrapure water or alkaline NaOH solution [59].

**Figure 10 materials-16-03979-f010:**
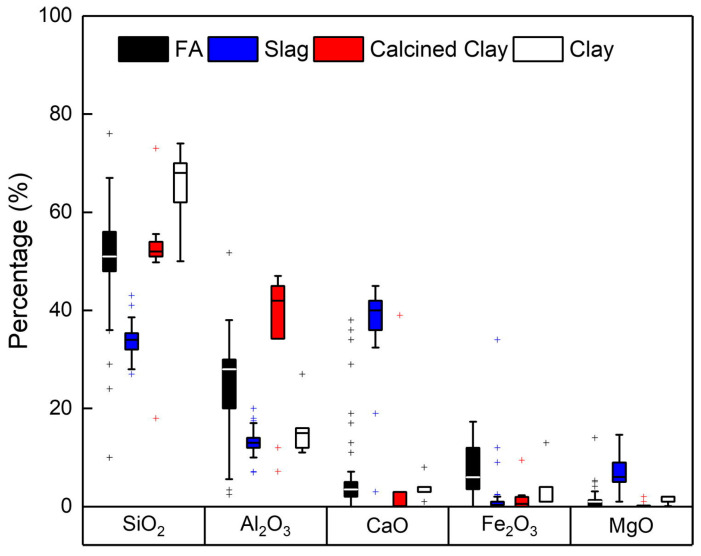
Average chemical composition of four aluminosilicate AAM precursors [42].

**Figure 11 materials-16-03979-f011:**
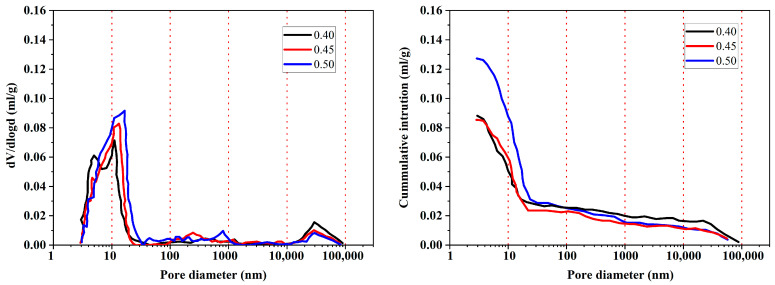
Effect of water-binder ratio on pore volume and pore size distribution [79].

**Figure 12 materials-16-03979-f012:**
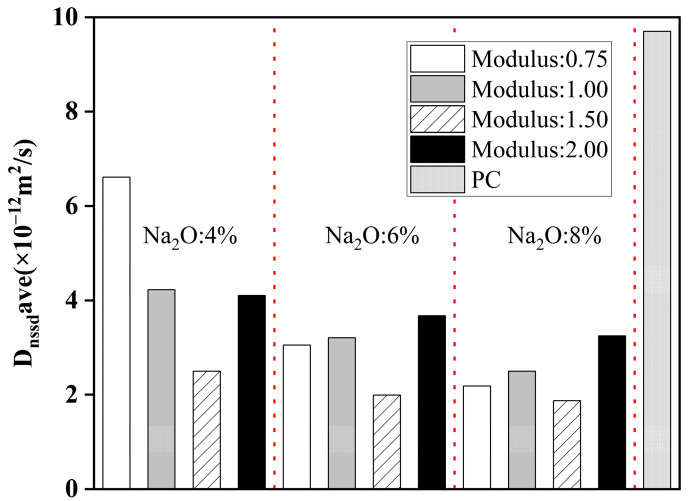
Non-steady state chloride diffusion coefficients of the concretes [119].

**Figure 13 materials-16-03979-f013:**
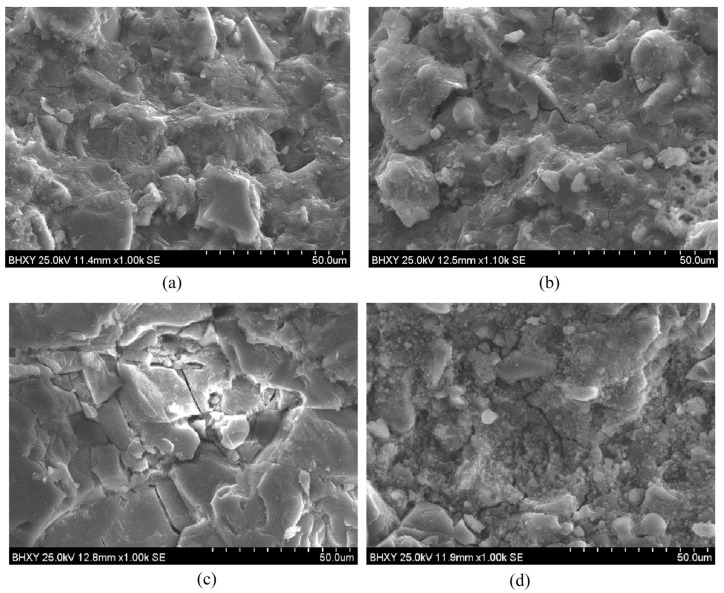
SEM images of (**a**) NaOH-activated slag, (**b**) NaOH-activated slag with 10% fly ash, (**c**) water glass-activated slag, and (**d**) water glass-activated slag with 10% fly ash [120].

**Table 1 materials-16-03979-t001:** Main element composition in the pore solution of AAFA, AAS, and OPC at 28 days (mmol/L).

Samples	Na	K	Ca	Al	Si	S	Reference
AAFA40 °C	1323	26.4	1.52	12.1	7.7	356	[51]
AAFA60 °C	792	9.8	0.99	2.0	25.7	317	[51]
AAS	1400	45	0.36	4.3	9.6	550	[53]
OPC	63	560	1.2	0.27	0.12	-	[53]

## Data Availability

Not applicable.

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
