# Peer review of "Chloride Transport and Related Influencing Factors of Alkali-Activated Materials: A Review"

_materials, 2023, doi:10.3390/ma16113979_

Round 1

Reviewer 1 Report

1.    The overall recommendation should be reported in one sentence at the end of the abstract.

2.   The authors should review the recent advancements made in the relevant field over the past two years.

3.   The authors have not extensively addressed the polymerization mechanism in geopolymer, which is a critical aspect of the paper. It is essential to comprehend the chemical reactions and polymerization process involved in geopolymer, as they govern the material's physical and mechanical properties. The following articles could serve as valuable articles for a more comprehensive and in-depth discussion of this topic: 10.1016/j.conbuildmat.2023.130688 ; 10.3390/polym15030615; 10.1016/j.jmrt.2023.02.088

4.  In Section 2.2, authors need to address some characteristic parameters defined according to pore size distribution. What techniques can be used to obtain pore structure characteristics? What is the role of capillary pores in chloride ion transport? How do different pore structure characteristics, such as total specific pore volume and pore diameter, correlate with the chloride diffusion coefficient in concrete?

5.  In Section 2.3, what are the differences in the composition and porosity of the interfacial transition zone (ITZ) between Portland cement (PC) concrete and alkali-activated materials (AAMs)? What is the main product found in the ITZ of alkali-activated fly ash concrete, and how does it differ from PC concrete? How is the ITZ characterised in geopolymer, and what features can be observed in the binder close to the aggregate?

6.   In Section 2.3, what are the outer products of ITZ in alkali-activated slag (AAS) concrete and how do they compare in terms of porosity with PC concrete? How does the evolution of ITZ in alkali-activated fly ash-slag concrete change over time, and what is the impact on porosity? How does the compact and dense microstructure of ITZ in alkali-activated concrete affect the transport of chloride ions and other aggressive ions, as mentioned in previous studies?

7.  In Section 3.1, how are chlorides adsorbed onto the surface of hydration products in alkali-activated materials (AAMs) due to intermolecular forces such as Van Der Waals and electrostatic interactions? What is the main mechanism of chloride solidification in AAMs, as observed in previous studies?

8.    In Section 3.1, how does the physical adsorption capacity of N-A-S-H gels, which are the main product in geopolymer, compare to other hydration products such as C-A-S-H gels? Are there any similarities between the microstructure of geopolymer and that of zeolite, which is known for its absorbent properties? How does the content of fly ash in alkali-activated slag affect the chloride binding capacity of N-A-S-H gels, as reported by Lee et al.?

9. In conclusion, it is recommended to provide a concise and comprehensive summary of the overall findings and conclusions in a single sentence, which should be placed at the end of the abstract to provide a clear and concise summary of the study's main recommendations.

Moderate editing of English language is required. 

Reviewer 2 Report

Chloride transport and related influencing factors of alkali-activated materials: A review

Abstract:

All components present are present:: objective/methods/ results/conclusions, future opportunities,

but may need minor writing for readability. The subject matter is original, press-worthy, of general interest

Remark:

Alkali-activated materials (AAMs) have been known for its environmental friendly features, it also behaves many excellent characteristics including acceptable durability.

This is a not completely true please comment, be careful when generalizing.

Introduction:

The introduction is well‐written and concise.

Hypothesis and purpose of study are clearly and concisely presented • Hypotheses are correctly presented and fully supported by text • Current references that will be of interest to readers, although many references are Chinees.

2. Chloride transport and micro-structure of AAMs

3. Chloride solidification in AAMs

4. Factors from fabrication of AAMs

5. Test method of chloride transport performance

General Remark

Description of procedures missing or so unclear that others couldn’t reproduce the study, with little likelihood that deficit could be fixed.

6. Conclusion:

Statements and conclusions are presented but need minor revision, to improve the clarity.

Minor editing is still needed

Reviewer 3 Report

The manuscript ID-2366644, titled “Chloride transport and related influencing factors of alkali-activated materials: A review” is a review paper that requires a dense revision and polishing before publication.

Abstract

This section required a precise work to deliver all content of the manuscript in the abstract.

Please provides of the work in the abstract.

Keywords: please use ; (semicolon between all words)

Introduction

This section is so short and should include more studies on AAM, please search for the same topic, there is many in google scholar.

The last paragraph, it should be revised carefully and discuss most of the work in there and mention on the future work that needs to be done later.

Chloride transport and micro-structure of AAMs

The authors talk on the pressure of the pore water in the concrete matrix, authors is best to check earlier that have been discussed on that such as “Pressure exerted on formwork by self-compacting concrete at early ages: A review”.  The authors should explain all detail of how it works.

Figure 3, it is not clear SEM images please provide high-resolution images to show all details.

Figure 5 (a) how is possible the 75% Fa+25% GGBS has a lower pore diameter compare to HPC is that mean suppose to have a higher strength in 75%FA and 25%GGBs.

Figure 6, it is a very low resolution. Provide a better images.

Figure 7 authors should use standard deviation to show the results on the graph for each of OPC, WAS, NAS.

IN PAGE 7 why the Figure 7 is colored red, please fix it.

Figure 11 the detail on the image is too small, please enlarge them to better present.

Figure 12 enlarges the x- and y- axes info.

Figure 13 enlarges the x- and y-axis info.  It is very small and hardly can see by the naked eye.

Figure 14 authors should show what the target to see these SEM images are. Please explain some clarification on it.

Conclusion

It is a bit long and confusing, please clarify the most important outcomes in this section. Please do not add any extra info.

Please use bullet points to show the most important outcomes. In the last paragraph explain what is necessary to do in future and what gap should be filled.

References

It is okay.

Proofreading is required

Round 2

Reviewer 1 Report

the revised manuscript can be accepted. 

Minor editing of English language required.

Reviewer 3 Report

The manuscript is now in the better edition. 

There is still a minor mistake, authors could revise them to improve the proofreading of the manuscript.